# Sustainable Waste Management in the Production of Medicinal and Aromatic Plants—A Systematic Review

**Sara Marcelino** [1], **Pedro Dinis Gaspar** [1] **and Arminda Paço** [2,*]

1   C-MAST—Centre for Mechanical and Aerospace Science and Technologies, Department of Electromechanical Engineering, University of Beira Interior, 6201-001 Covilhã, Portugal; sara.marcelino@ubi.pt (S.M.); dinis@ubi.pt (P.D.G.)
2   NECE—Research Unit in Business Sciences, Department of Management and Economics, University of Beira Interior, 6201-001 Covilhã, Portugal
*   Correspondence: apaco@ubi.pt

**Abstract:** Without a Sustainable Waste Management (SWM) system, the growing demand for Medicinal and Aromatic Plants (MAPs) can also lead to a considerable increase in the waste generated by the industry. Since MAP residues have a notable potential to be valorised, the implementation of Circular Economy (CE) solutions can play a central role in converting waste into economic opportunities, while fostering a sustainable planet. CE helps to mitigate environmental and social risks caused by the accumulation of biomass by turning waste into valuable products. A systematic review was conducted, aiming to identify potential applications for the valorisation of MAP residues under a sustainable approach. A total number of 47 studies were analysed, providing a novel compilation of possibilities for decision makers in the MAP industry to develop new products for crop management or new businesses in food, cosmetic, pharmaceutical, chemical, paper, or building industries. Researchers in this field have focused more on the industrial value of MAP residues than on the empirical assessment of environmental and economic benefits. Further investigation should be undertaken to present empirical applications and to develop a decision support system to assess the sustainable performance of valorisation options.

**Keywords:** waste management; medicinal and aromatic plants; circular economy; sustainability; valorisation

## 1. Introduction

There has been a growing interest in natural plant-derived products [1]. According to the International Trade Centre [2], the world trade of Medicinal and Aromatic Plants (MAPs) resources is growing at the rate of 10–12% annually. With the rising demand for MAPs, the risk of overexploitation increases [3]. Overexploitation can generate changes in the structures and population dynamics patterns of extracted species. Once overexploitation affects plants' growth and reproductive capacity [4], it may even lead to the loss of existing populations [5]. However, MAP overharvesting does not only generate harmful effects for these individual species; it can also adversely affect the wildlife population and result in the loss of multiple ecosystem services [3]. Furthermore, along with the increase in the world trade of MAPs, the waste generated in the industry also increases, which in turn can foster the negative effects of inadequate waste management. It is estimated that the medicinal herb industry produces nearly 30 million tons of waste every year [6]. Despite containing valuable substances, the massive herb residues are usually discarded through stacking, landfill or burning, which is not only a waste of resources but also causes considerable environmental pollution [7].

At a global level, the amount of residues generated is also increasing, incrementing the risks associated with improper waste management, which are both environmental and social [8,9]. On an environmental level, it results in groundwater pollution, clogging of

drains and the emanation of air pollutants [10,11]. The landfilling of generic solid waste is responsible for 2% of global greenhouse gases and emissions triggering climate change [9], such as methane ($CH_4$), biogenic carbon dioxide ($CO_2$) and non-methane volatile organic compounds (NMVOCs) [12]. It is estimated that the emission of methane (a gas with a global warming potential 25 times greater than carbon dioxide) from landfills accounts for 3–9% of the anthropogenic sources in the world [12]. Moreover, the concentration of atmospheric methane is annually increasing by 1–2% [12].

Apart from the environmental hazards, inadequate waste management leads to public health risks, especially for people living near the open dumpsites, due to spontaneous fires and toxic gas emissions. It also promotes the dissemination of diseases like diarrhoea, dysentery, and eczema, among other socio-economic problems [11,12].

Particularly, the biomass accumulation from MAPs, especially in regions with dry climates, leads to fires, and consequently to large losses of forests and biomes. Additionally, on a large scale, biomass accumulation contributes to the production of greenhouse gases and the proliferation of pathogenic microorganisms [13]. Furthermore, the incineration of agro-industrial wastes is mainly associated with the production of greenhouse gases, leading to the loss of energy potential and affecting the population's health and quality of life [13].

Since the increase in MAP demand intensifies the risks associated with overexploitation and inadequate waste management, it is urgent to implement sustainable practices. However, according to Taghouti et al. [14], the collectors' lack of information about sustainable management is one of the weaknesses of the most productive countries around the Mediterranean MAP sector. Concerning the transformation and processing, the same study highlights the valorisation of the by-products obtained in the distillation and drying processes as one of the faced challenges by the sector [14]. Although MAP residues are recognised as being a potential source of valuable compounds, there is an evident gap between the theoretical knowledge and the application of sustainable waste management approaches by the industry.

Circular economy is viewed as a solution for fostering a sustainable system [15] and aims at the elimination of waste [16]. Thus, circular economy can be defined as the transition from a linear model, where resources are transformed, used, and discarded, to a circular (regenerative) model in which materials are reused whenever possible [17]. Circular Economy solutions are aligned with the United Nations Sustainable Development Goals requiring transformation of the current business practices [18]. Additionally, applying Circular Economy strategies can be economically beneficial and it can also be a practical solution for the optimum use of biological resources, encouraging more viable and responsible sustainability strategies in the MAP sector [1,5]. Furthermore, through sustainable waste management approaches in agriculture, and particularly in MAP production, it is possible to explore valuable natural resources generating economic value at a low environmental cost, without threatening the accessibility and availability of plants, as well as the general well-being of future generations [19].

In this study, a systematic review was conducted, aiming to answer the following question: How can MAP production residues be valorised under a sustainable waste management and a circular economy approach? It is also intended to clarify what the sustainable benefits of waste valorisation applications are. Thus, this study makes a novel contribution by providing a set of possibilities for residue valorisation in the MAP sector, that were previously scattered in the literature. Additionally, topics are presented for future research based on the recent literature of this growing interest research field.

This article starts with a description of the methods applied to extract the analysed literature. Section 3 presents the descriptive and thematic results, describing the variety of waste valorisation methods for MAP residues presented in literature and some positive effects associated. The discussion section explains the main findings, limitations, and topics for future investigation and, in conclusion, the possible applications for MAP residues are summarised.

## 2. Materials and Methods

The systematic literature review was performed following the PRISMA (Preferred Reporting Items for Systematic Reviews and Meta-analyses) guidelines, which were conceived to promote a transparent report of results [20]. The PRISMA methodology is divided into four stages: identification, screening, eligibility, and inclusion for analysis [21].

In April 2023, identification, i.e., data collection for this systematic review, was carried out. Three electronic databases were consulted, namely, Scopus, Web of Science, and PubMed. Table 1 exhibits the keywords chosen to conduct the search. Since it was verified that using only one search equation significantly limited the number of results, four different search equations were used. No time restriction was considered.

**Table 1.** Keywords and search results for the systematic literature review.

| Search String | Database | Articles Found |
|---|---|---|
| TITLE-ABS-KEY ("circular economy" OR "circular farming" OR "circular agriculture") AND ("aromatic plant*" OR "medicinal plant*" OR "aromatic herb*" OR "medicinal herb*" OR "aromatic flower*" OR "medicinal flower*") | Scopus | 21 |
| | Web of Science | 18 |
| | PubMed | 8 |
| TITLE-ABS-KEY ("*waste* management" OR "*waste* valorization" OR "*waste* valorisation" OR "*waste* reuse" OR "*waste* recycling" OR "management of *waste*" OR "valorization of *waste*" OR "valorisation of *waste*" OR "reuse of *waste*" OR "recycling of *waste*") AND ("aromatic plant*" OR "medicinal plant*" OR "aromatic herb*" OR "medicinal herb*" OR "aromatic flower*" OR "medicinal flower*") | Scopus | 59 |
| | Web of Science | 24 |
| | PubMed | 14 |
| TITLE-ABS-KEY ("*residue* management" OR "*residue* valorization" OR "*residue* valorisation" OR "*residue* reuse" OR "*residue* recycling" OR "management of *residue*" OR "valorization of *residue*" OR "valorisation of *residue*" OR "reuse of *residue*" OR "recycling of *residue*") AND ("aromatic plant*" OR "medicinal plant*" OR "aromatic herb*" OR "medicinal herb*" OR "aromatic flower*" OR "medicinal flower*") | Scopus | 4 |
| | Web of Science | 5 |
| | PubMed | 122 |
| TITLE-ABS-KEY ("by-product* management" OR "by-product* valorization" OR "by-product* valorisation" OR "by-product* reuse" OR "by-product* recycling" OR "management of by-product*" OR "valorization of by-product*" OR "valorisation of by-product*" OR "reuse of by-product*" OR "recycling of by-product*") AND ("aromatic plant*" OR "medicinal plant*" OR "aromatic herb*" OR "medicinal herb*" OR "aromatic flower*" OR "medicinal flower*") | Scopus | 3 |
| | Web of Science | 1 |
| | PubMed | 21 |
| Total | | 300 |

The 300 articles initially obtained were manually screened based on the title and abstract reading and the data was collected by two reviewers. Articles were excluded if:

- There was a duplicate article in another database;
- The full text could not be accessed;
- They were not written in English;
- They were not related to MAPs;
- They did not refer to the circular economy or waste management concepts.

In the eligibility phase, articles were excluded if:

- They were not related to MAP residues (Reason 1—Figure 1);
- They did not present potential applications to valorise MAP residues (Reason 2—Figure 1).

A PRISMA flowchart illustrates the step-by-step process of the selection of studies, using specific inclusion and exclusion criteria to generate a final number of studies for analysis [21]. Figure 1 was elaborated according to the PRISMA 2020 statement revised flow diagram [20].

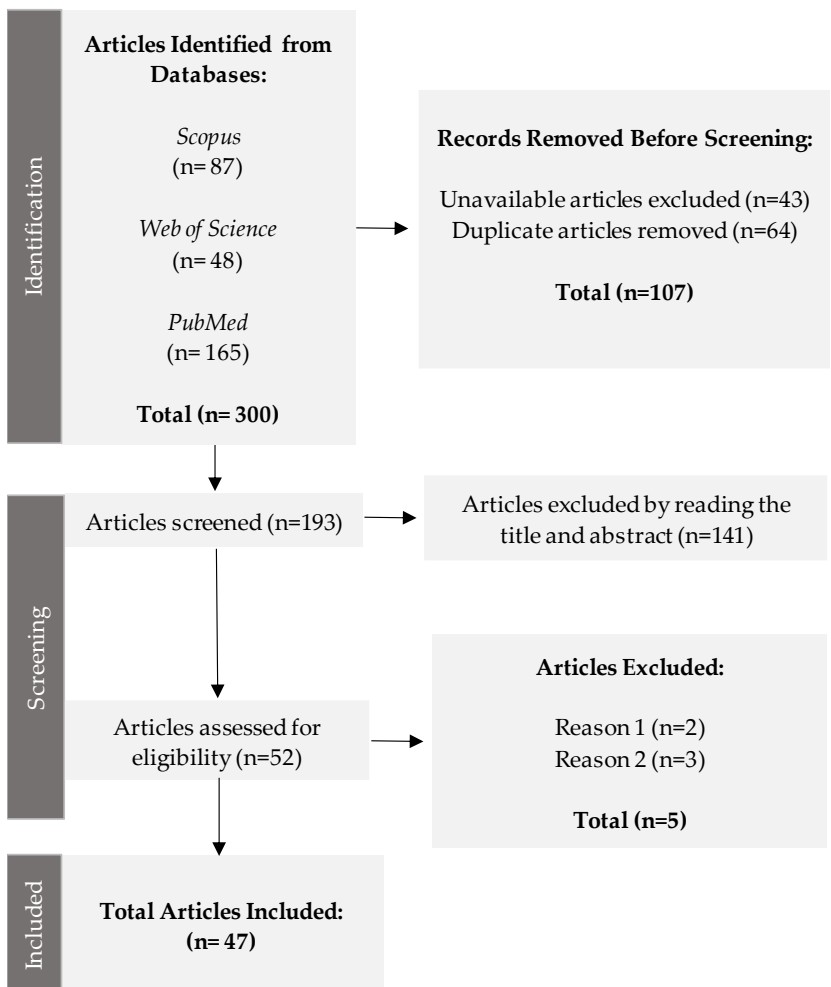

**Figure 1.** Study selection.

## 3. Results

*3.1. Descriptive Results*

The analysis of the extracted literature concerning the number of articles published by year allows us to conclude that from 2019 until 2022 the number of published articles increased each year, which is evidence of the growing research interest in waste valorisation solutions for the MAP sector (Figure 2).

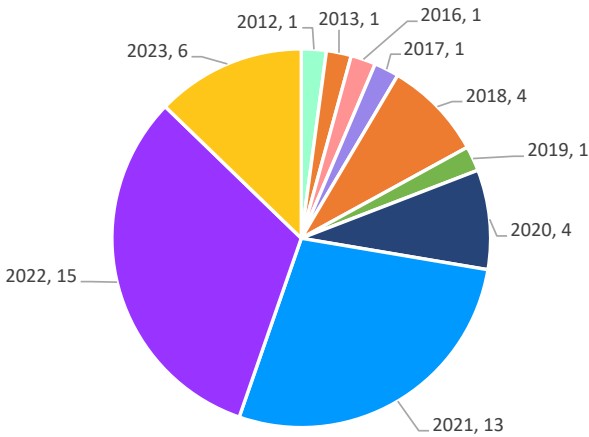

**Figure 2.** Number of articles published by year.

The locations shown in Figure 3 correspond to the origin associated with the first author or the respective affiliation. It was verified that China is the country with the highest number of publications (10). This feature is congruent with the considerable use of plants in Traditional Chinese Medicine practices, and the consequent interest in the efficient use of these natural resources. Next to China, Italy, France, and India are the countries with a higher number of publications. This information allows us to identify for the first-time what countries contributed more to the literature on waste valorisation in the MAP sector.

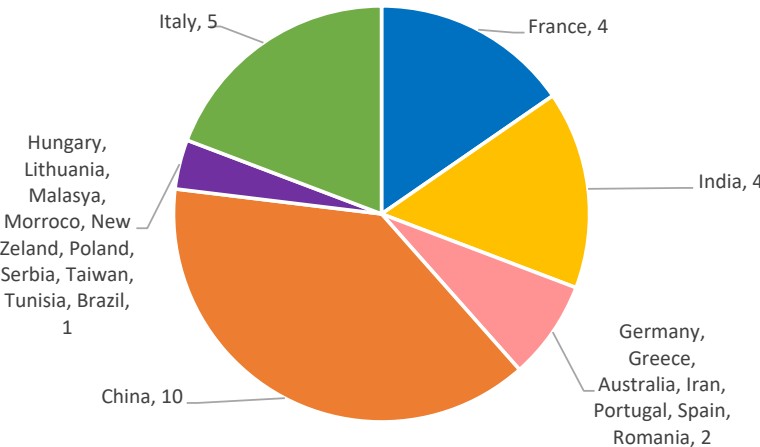

**Figure 3.** Number of articles published by location.

*3.2. Thematic Results*

The possible solutions to valorise MAP residues, converting them into value-added goods, are presented in this subheading. These solutions include energy purposes, crop management and applications in chemical, food, pharmaceutical, and cosmetic industries, among others.

3.2.1. Energy Purposes

Herb residues can be converted into gas fuel to provide energy for productive systems. For instance, Guo et al. [22] designed a method to valorise herb residues through gasification for industrial-scale utilisation. It is also proposed to use the heat of the flue gas to recycle energy when drying herb residues.

Plant solid residues obtained from hydrodistillation can be used for energy production through direct burning, making briquettes after drying, pelletising and gasification. Indirectly, these residues can be used in biomethanisation and pyrolysis processes, to obtain biodiesel and biochar for energy purposes [23,24].

According to Robertson et al. [25], cannabis waste, including cannabis biomass, solvents, packaging and unused products, can be used for incineration, i.e., to recover energy exposed from the combustion of waste in a furnace and reduce the volume and mass of waste generated.

Beyond incineration, cannabis waste can be used for anaerobic digestion. In anaerobic digestion, microorganisms digest the waste in the absence of oxygen, which generates a digestate and biogas. The produced biogas can be used for sustainable energy production [25].

Fardad et al. [26] presented the results from mesophilic anaerobic digestion of waste from the following species: *Glycyrrhiza glabra*, *Mentha*, *Cuminum cyminum*, lavender and Arctium. The study concluded that through anaerobic digestion, these residues allow to produce biogas and to recover methane as a source of renewable energy, instead of burning or burying them [26].

Although no study referred to any assessment methods to choose an appropriate option for the sustainable use of MAP residues for energy purposes, the literature analysis allows us to point out some relevant variables that must be considered, namely the emission

of greenhouse gases, the fuel required to initiate the combustion reaction in incineration, the obtained reduction of solid waste, and energy savings [25].

### 3.2.2. Crop Management

MAP by-products may be rich in bioactive compounds of interest and beneficial microorganisms. Hence, they can be used as natural biopesticides or organic fertilisers, which may suppress the growth of soil-borne phytopathogens [27]. Under a circular economy approach, the use of these residues is also recommended for the restoration of degraded and marginal lands [23]. Solid residues from distillation also exhibited the capacity to stimulate seed growth [28].

Through composting and vermicomposting technologies, these by-products can be used to improve relevant soil processes and provide nutrients [27]. Composting has become a very important sustainable approach to recycle medicinal herbal residues [29] and it allows the conversion of organic waste into stabilised soil amendments, under aerobic conditions [25].

Zaccardelli et al. [24] demonstrated that through the combination of different techniques, it is possible to valorise all fresh aromatic plant residues, where composting is one of the applied techniques. The study used residues from cropping and processing activities, namely the whole basil plants at the end of the cycle and the pruning residues from rosemary, as well as all the green waste of basil, rosemary, and sage, including leaves, stems and inflorescences. The oil-free biomasses from the harvesting, production, and processing of MAPs are suitable for use in agriculture for composting and to obtain a high-quality organic fertiliser [24].

Regarding the use of antibiotic-contaminated swine manure in agriculture, the study of Zhang et al. [30] suggested that anaerobic co-digestion with Chinese medicinal herbal residues could be employed to remove some antibiotic resistance genes (considered emerging pollutants harmful to the environment and human health) and mobile genetic elements from swine manure.

Like composting, the process of anaerobic digestion, using herb residues, generates a digestate that can be used as a fertiliser since it is high in minerals and nutrients beneficial for plant growth [25].

The study of Wei et al. [31] used traditional Chinese medicinal herb residue to produce biochar adsorbent by pyrolysis. The use of biochar as a soil amendment can boost soil productivity and mitigate climate change, and it is also indicated as a sorbent for immobilisation of potentially toxic elements and organic contaminants in the soil, water, and air.

Basak et al. [32] described a novel biochar–mineralcomplex, which was made using the distillation waste of lemongrass (*Cymbopogon flexuosus*). The study found that the biochar–mineral complex enhanced the soil quality by improving available nutrients and biological activities. Thus, the use of waste of lemongrass can be incorporated in biochar–mineral complex formulations to produce an excellent starter fertiliser, as an alternative to chemical fertilisers [1].

Regarding the effectiveness of herb residues as organic fertilisers, the study of Ma et al. [33] revealed that Chinese medicinal herbal residues can replace 25–50% of the typically applied quantity of chemical fertiliser in fields, without hampering maize grain yields. Such application allows the recycling of waste, the improvement of soil fertility and the decrease of chemical fertiliser usage. Additionally, the research performed by Filipović et al. [34] confirmed that using compost from organic waste, obtained in the processing of medicinal plants, specifically on marigolds, enhanced the productivity and quality of plants as if they were fertilised with conventional fertiliser. These results suggest that MAP waste can be successfully applied as an alternative to other organic and chemical fertilisers [34].

Using MAP waste biomass to obtain soil amendments, such as compost and biochar, has the potential not only to mitigate greenhouse gases but also, at least partially, to replace chemical fertilisers. Furthermore, the application of biochar improves the physical, chemical, and biological properties of soils, supplying and retaining nutrients [1].

The study of Perlein et al. [35] about the characteristics of MAPs cultivated in contaminated lands demonstrated that sage (*Salvia sclarea* L.), presenting an excluder phenotype, is suitable for phytostabilisation, and the waste from extraction can be applied as a soil amendment. On the other hand, coriander (*Coriandrum sativum* L.) distillation residues showed relatively elevated Cd concentrations, which limits their utilisation [35].

The study by Fontaine et al. [36] also indicates that, although *C. sativum* is useful for the phytomanagement of polluted soils by trace elements, the valorisation of its distillation residues is limited since it is a cadmium (Cd) accumulator, with Cd being recognised as one of the most dangerous pollutants because of its severe toxicity and high mobility in soil.

Beyond all the previously mentioned possible uses for residues from MAPs production, solid residues from distillation can also be used as biosorbents, as soil organic cover or for weed management. Also, the hydrolates can offer a source of active compounds to profitably make eco-friendly biopesticides [24,28].

In summary, it can be pointed out that utilising MAP residues can directly benefit crop management. Some recent studies [33,34] highlight the potential of organic fertilisers from MAP residue to entirely or partially replace chemical fertilisers. However, more studies are needed to test the efficacy and efficiency of organic fertilisers on a large scale.

### 3.2.3. Chemical Industry

MAP biomass can also be a feedstock for the green chemistry industry, allowing it to obtain valuable compounds that can be used, for instance, in medicinal applications.

Kong et al. [7] proposed the valorisation of Chinese herb residues combining endophytic and probiotic fungus *Aspergillus cristatus* CB10002 to produce anthraquinones, an important chemical compound with diverse medicinal properties.

MAP residues can also be applied to enzyme synthesis. Through the delignified bioprocessing of MAP residues, such as Java citronella (*Cymbopogon winterianus*) and Artemisia (*Artemisia annua*), it is possible to produce cellulase enzymes [1].

Also, the study by Lesage-Meessen et al. [37] indicated that the distilled straws of lavender and lavandin are raw materials of interest for producing platform molecules (e.g., antioxidants) and fungal enzymes involved in the decomposition of recalcitrant lignocellulose into biofuel.

These approaches allow high-added value compounds to be obtained from cheap and easily available by-products while fostering the development of a circular bioeconomy [37].

It was verified that the studies regarding MAP residue potential in the chemical industry are mainly focused on descriptive analysis of their properties. More studies should be conducted to verify the environmental and economic advantages and efficacy of such applications in a real context.

### 3.2.4. Food Industry

MAP by-products can be valuable for the food industry since they are natural sources of functional ingredients with high nutritional value and bioactive compounds and their properties can be beneficial for animal or human consumption [38].

Some of the residues generated in the extraction of medicinal herbs, especially when ethanol or water are used as solvents, can be used as feed additives in livestock farming, since they still contain 30–50% essential bioactive compounds [6].

A review article about the application of medicinal herbs in aquaculture revealed that the combination of probiotics and medicinal herbs as feed additives improves the aquaculture species' growth performance, immune system, and disease resistance. Thus, the use of MAP residues can be useful for aquaculture industry [6].

Andreadis et al. [38] conducted a study about the use of post-distillation residues of lavender, Greek oregano, rosemary, and olive (in equal proportions), for substrate supplementation with several agricultural residues. The results showed that MAP residues had a favourable effect on the total phenolic content and antioxidant activity of each substrate. Such results indicate that alternative substrates can be developed and their enrichment

using MAP residues can influence the growth of T. molitor (an important commercial edible insect particularly valued for its protein content). Moreover, solid residues from MAP hydrodistillation can be considered as a low-cost deodorised material, free of volatile compounds. Since odorous ingredients can impair the organoleptic properties of feed products, or even be rejected when used in animal feed, these solid residues have a useful potential in the food industry, in both animal feed and food supplements [28]. For instance, the solid residue from rosemary (*R. officinalis*) has the potential to be used as an antioxidant feed supplement for pregnant ewes and it can reduce lipid oxidation of meats. This solid residue is even considered by the European Commission as a semi-natural additive for the conservation of food [28]. The use of ethanol extracts from the solid residues of lavender (*Lavandula angustifolia*) and lemongrass (*Melissa officinalis*) is also suggested to extend the shelf life of bread [28]. Furthermore, extracts of residual water (i.e., the water that came into contact with the aromatic herb in the distillation process) can be applied as nutraceutical, functional food additives and food antioxidants [28].

Antioxidant properties are demanded not only in pharmaceutical and cosmetic industries, but also in the food sector, since many processed products require the addition of stabilising, colouring, or preserving ingredients. Due to the growing interest in preparations without synthetic additives, plant extracts are increasingly included as natural antioxidant compounds in food industry formulations [39]. In some cases, deodorised plant extracts obtained from the oil-free biomass exhibit higher antioxidant properties than the extracts isolated from the whole material. Hence, since some MAP residues exhibit significant antioxidant activity, there is potential to increase their use in the food sector [40].

Concerning the food industry's applications, there are some products available on the market with residue extracts [41] and the potential of some residues is even recognised by the European Commission. However, there is evidence of a wide range of possibilities that are yet to be explored empirically, as suggested by the information provided in Table 2. Table 2 summarises the residue reuse applications of specific MAP species presented in the extracted literature, including phytochemical studies and review articles.

**Table 2.** Properties and potential applications for MAP residues.

| Plant Specie | Origin of Sample Materials | Waste Type | Properties | Application | Reference |
|---|---|---|---|---|---|
| Mangaba (*Hancornia speciosa* Gomes, Apocynaceae) | North-east Brazil | Leaves | Extract rich in cyclitols and flavonoids and high amounts of bornesitol. | Development of antihypertensive herbal medicine or as source of the bioactive constituent bornesitol. | [17] * |
| Caryocar brasiliense A.St.-Hil., (Pequi) | North-east Brazil | Peels | Tannins, including corilagin and geraniin; antiviral properties. | Development of antidiabetic and antiviral herbal medicine. | [17] * |
| Chamomile (*Matricaria recutita* L. and *Matricaria discoidea* DC) | Germany | Roots of flower production for tea | Middle polar extracts have bioactive phytochemicals, potent antioxidant, and antibacterial activity. | Phytomedicinal or cosmetic preparations (oil-based cosmetic products) | [39] |
| *Calamintha grandiflora* L. | South-west of France | Water extracts | Antioxidant activity and offers protection against oxidative deterioration. | Food industry, including the formulation of food additives and healthy supplements. | [40] |
| | | Solid hydrodistillation residue | Antioxidants and volatile aroma compounds. | | |
| Pine (*Pinus pinaster*) | - | Shoots | Rich source of natural polyphenols. | Pharmacological applications. Production of pine shoot syrup, pine shoot-based beer and herbal teas. | [41] * |
| | | Bark extracts | Photoprotective and anti-photoaging activities. | Pharmaceutical industry. | |
| | | | Antioxidant capacities. | Food applications to extend shelf-life. | |

**Table 2.** *Cont.*

| Plant Specie | Origin of Sample Materials | Waste Type | Properties | Application | Reference |
|---|---|---|---|---|---|
| Argan (*Argania spinosa* L.) | South-western Morocco | Shell fruit from argan oil production | Ethanol extract has high level of total phenol content, flavonoids, condensed tannins, and flavanol; has a potential antioxidant effect, potential anti-inflammatory, and antioxidant activities. | Pharmaceutical and food industries. | [42] |
| Lotus (*Nelumbo nucifera* Gaertn.) | - | Lotus seedpods | Extracts exhibit antioxidant, anti-cancer, anti-melanogenic, anti-inflammatory, anti-irradiation, cardioprotection and hepatoprotection activities. Water extracts exhibit antioxidation, anti-cancer, anti-melanogenic, anti-inflammatory and hepatoprotection activities. | Health food and pharmaceutical industry. | [43] * |
| Lovage (*Levisticum officinale* W.D.J. Koch) | Spain | Dried Roots | Bioactive compounds, such as phthalides and phenolic acids. | Food and/or the pharmaceutical industry. | [44] |
| Elderberry (*Sambucus nigra* L.) | Spain | Hydrolates | Active principles of elderberry bark and stems. | Cosmetic industry, as skin tonic or for therapeutic baths. | [45] |
| Rosemary (*Rosmarinus officinalis* L.), Greek sage (*Salvia fruticosa* L.) and Spearmint (*Mentha spicata* L.) | Greece | Distilled solid residues | Rich in bioactive compounds with antioxidant activity, mainly polyphenols. | Food, pharmaceutical and cosmetic industry. | [46] |
| Oregano (*Origanum vulgare* var. aureum), Thyme (*Thymus vulgaris*, var. Doone Valley) and Summer savory (*Satureja hortensis*) | Romania | Distilled residues | Hydro-alcoholic extracts have high phenolic content which is a valuable source of biologically active compounds, such as antioxidants. | Food and pharmaceutical industry. | [47] |

**Table 2.** *Cont.*

| Plant Specie | Origin of Sample Materials | Waste Type | Properties | Application | Reference |
|---|---|---|---|---|---|
| Tobacco (*Nicotiana tabacum*) | - | Tobacco processing waste | Extracts exhibit anti-inflammatory, antitumor, antibacterial, and antioxidant functions. | Food additive, cosmetic and pharmaceutical industry to provide resistance to numerous diseases, regulation of human health, sterilisation, and pest control. | [48] * |
| *Trichodesma khasianum* and *Euphorbia hirta* | Taiwan | Euphorbia hirta aerial parts Trichodesma khasianum leaves | Bioenergy potentials for antiviral activities. | COVID-19 drug development. | [49] |
| Lemongrass (*Cymbopogon citratus*) | Portugal | Hydrolate | Composed of emulsified citral-rich essential oil. | Functional ingredient in a *matcha* tea formulation to provide taste and extended shelf life. | [50] |
| | | Non-distilled aqueous phase (decoction) | Glucose-rich polysaccharides, and antioxidant and anti-inflammatory properties. | Possible application as a functional dietary fibre in the food industry. | |
| Lavender (*Lavandula angustifolia* Mill. (LA) and *Lavandula* × *intermedia* Emeric ex Loisel (LI)) | Italy | Oil-exhausted biomasses from distillation | Antioxidant and anti-tyrosinase activities. | Food and cosmetic industries to prevent the browning and deterioration of active compounds and improve the conservation of final products. | [51] |
| | | | Anti-enzymatic capabilities. | Pharmaceutical industry to create a therapeutic alternative for the prevention and treatment of chronic diseases such as Alzheimer's disease and hyperpigmentation. | |

**Table 2.** *Cont.*

| Plant Specie | Origin of Sample Materials | Waste Type | Properties | Application | Reference |
|---|---|---|---|---|---|
| Rosemary (*Salvia rosmarinus* Schleid., formerly *Rosmarinus officinalis* L.) | Germany | Post-distillation residual water, spent plant material extracts, and post-supercritical $CO_2$ spent plant material extracts | Antimicrobial, antioxidant, and enzyme-inhibitory activities. | Pharmaceutical, cosmetic, and nutraceutical industries. Terpene-rich extracts can be used as food preservatives (antioxidants) or aroma-active ingredients. | [52] |
| *Cymbopogon winterianus* Jowitt (Java citronella) | India | Solid distillation waste | Antioxidant activity. | Dietary industry. | [53] |
| Thyme (*Thymbra capitata* L.) | Tunisia | Post-distilled residues | Antioxidant, antimicrobial, anti-biofilm, anti-inflammatory and anticarcinogenic capacities. | Replace or even decrease synthetic antioxidants in foods, cosmetics, and pharmaceutical products. | [54] |
| Ginger (*Zingiber officinale*) | Italy | Lees | Extracts have an appreciable amount of oleoresin rich in gingerol-like compounds. | Flavour, perfume, and nutraceutical sectors. | [55] |
| Chicory (*Cichorium intybus* L.) | - | Forced chicory roots | Caffeoylquinic acid and antioxidant activity. | Possible therapeutic application. | [56] * |
| Rose (*Rosa damascena* Mill) | Iran | Residual water of hydrodistillation | Phenolic content (phenol, flavonoid, and anthocyanin). Recovered fraction of waste materials could be considered as a proper antioxidant, DNA damage-protection agent, and xanthine oxidase inhibitor. | Pharmaceutical and nutraceutical industry. | [57] |

* Article review.

### 3.2.5. Pharmaceutical, Natural Medicine and Cosmetic Industries

The guidelines of the European Commission validate the reuse of several types of organic waste [58] and there is evidence that MAP residue, in particular, has a considerable potential to be valorised. At the same time, the growing demand for bioactive compounds from MAP by pharmaceutical and cosmetic industries promotes the development of new methods that allow the sustainable and efficient exploitation of these natural resources [58].

An efficient valorisation of MAPs, including their residues, must consider, whenever possible, all their valuable properties and choose applications accordingly.

If herb residues still contain nutritional and medicinal ingredients, approaches such as pyrolysis, gasification to obtain fuel gas, or biochar production, among others, ignore the effective potential of materials and usually involve high energy consumption. Despite using MAP residues, such applications may not be the best option to foster environmental sustainability and to efficiently reuse plant materials [7].

For instance, solid residues obtained after extraction are mostly used for mulching or energy purposes. Still, such residues are rich in polyphenols that are not collected in the essential oil as they are not volatile [51].

Residues from the distillation of MAPs (solid residues, wastewater, and hydrolates) contain phenolic compounds of interest [38], such as terpenes, phenolic monoterpenes, phenolic diterpenes, hydroxybenzenes, phenylpropanoids, and flavonoids [28].

Flavonoids, which are present in some aromatic plant distillates, benefit the prevention of photoaging damage. The potential beneficial dermatological effects provided by these phenolic compounds present in MAP production residues may favour their application in modern medicine [58].

The study of Marzorati et al. [59] revealed the significant presence of unextracted compounds from *Cucurbita pepo* L. seeds and *Serenoa repens* L. fruits, such as fatty acids, sterols and polyprenols. Bioactive molecules from MAP residues can be directly usable as phytotherapic drugs or in combination with other ingredients in specific pharmaceutical formulations [59].

As many active ingredients remain after the extraction of essential oils, re-extraction is an attractive and necessary method to efficiently increase the exploitation of MAP properties and increase their economic value, while reducing waste [60].

The study of Meng et al. [61] presented another technique to valorise traditional Chinese herb residues in the health sector. It was concluded that herb residue fermentation supernatant successfully inhibited urease activity, reduced the level of some gastric inflammatory cytokines, alleviated histological lesions, and helped to retrieve the disturbed microbiota to a normal level. This finding demonstrates the potential of traditional Chinese herb residues and it can boost the development of innovative healthcare products [61]. Table 2 presents a summary of other relevant studies regarding potential applications in the pharmaceutical, natural medicine, and cosmetic industries.

The literature analysed in this section focuses on the potential of MAP residue without presenting information on the real impact that their utilisation can have on health and cosmetic product users. The number of commercialised products with MAP residues is still limited since their potential is underexploited. Studies on finding more efficient methods to extract compounds of interest to prepare new products are needed. With more knowledge about the efficacy and efficiency of MAP residue-based preparations, a gradual implementation by pharmaceutical and cosmetic industries is expected.

Furthermore, the health-beneficial properties of MAP residues can also be explored in the natural medicine industry to design new treatments, given the growing interest in natural plant-derived products [1]. Also, in this industry, more investigation is required for the assessment of the treatments' efficacy.

### 3.2.6. Further Applications

MAP residues can be valorised as raw material for paper-making [7] and in the wood-based panel industry [62].

Following the principles of circular economy, Fehrmann et al. [62] suggested that residual hemp (*Cannabis sativa* L.) biomass, namely hemp hurd, can be used in the manufacture of engineered lightweight panel products as an alternative renewable lignocellulosic feedstock. This strategy not only reduces waste production but also allows the rising resource shortage in the wood-based panel industry to be addressed.

Also, the residual biomass of MAPs can be used as an effective adsorbent for heavy metal and pigment removal and wastewater treatment [1].

Huang et al. [60] investigated bio-based materials production using traditional Chinese medicinal herb residues. Using *Radix bupleuri*, astragalus roots, *Radix liquiritiae*, *Fructus aurantii*, *Fructus gardeniae*, *Evodia rutaecarpa*, *Herba plantaginis*, lobelia, and *Cacumen platyclade* (namely rhizomes, fruits, and leaves, with 5:4:2 being the optimal ratio), it is possible to produce activated carbon for an efficient cephalosporin antibiotic adsorption in wastewater. The study demonstrated that low concentrations of both cephalexin and cefradine can successfully be adsorbed. It was also stated that, after ultrasonic treatment in water, the produced activated carbon can be reused for adsorption. Additionally, Chinese medicinal herb residues can be used for materials and chemical production, namely composites synthesis. The composite generated from *H. angustifolia* root xylem and polylactic acid exhibit promising comprehensive mechanical properties.

Another study indicates that Magnolia (*Magnolia officinalis*) bark extract residues due to their moisture and lignocellulosic contents can be a raw material to make composites which in turn can be adapted into products. The findings reveal that composites with 20% Magnolia bark extract residues in their composition resulted in products with optimal impact and bending strengths [63].

The study of Ratiarisoa et al. [64] used raw lavender straw, a mix of stems, flowers, and other degraded elements, to produce and evaluate a bioaggregate for building materials. It was concluded that, although further investigation is needed, distilled lavender straw can potentially be used to build bioaggregate, since the produced composite exhibited promising thermal and hygric performances. Nevertheless, lavender-based composite exhibited very weak mechanical performances and further tests should be carried out to identify an efficient method to valorise this specific type of residues for building applications.

Thus, MAP residues have the potential to be used in the production of paper, the wood-based panel industry, composites synthesis, wastewater treatment, and building materials. In the mentioned applications more studies are needed to develop efficient methods to valorise MAP residues and to assess sustainable benefits both economic and environmental.

### 3.3. Positive Effects of Waste Valorisation

Beyond the tangible benefits of waste valorisation from the MAP industry, there are sustainable benefits involving environmental, health and economic dimensions.

For example, when the landfilling or incineration of residues is replaced by the production of biochar, the greenhouse gas ($CO_2$) emission is reduced. Moreover, it was demonstrated that the use of biochar reduces the $NO_2$ emission from agricultural soils [1].

Applying MAP residues as soil amendments reduces the energy consumption in the production of agrochemicals and minimises environmental and health implications associated with chemical fertilisers use in agriculture [1].

The valorisation of MAP residues contributes to waste minimisation, climate change mitigation, soil carbon sequestration, soil fixation, and soil quality improvement, and provides a range of ecosystem services [23].

Under a circular economy approach, the organic waste of the MAP industry can be converted into economic opportunities. The implementation of circular economy strategies can play a central role in transforming the negative impacts of conventional waste management into economic opportunities and in responding to the growing demand for bioactive compounds in the pharmaceutical and cosmetic industries [58].

The use of MAP residues to create add-value products requires low-cost raw materials, and it can represent a business strategy that is green, economically viable, and scalable to industrial use [63]. Also, if cheap raw materials, such as agricultural residues, are used in enzyme production and other MAP-based products, production costs can be reduced and it would solve, to some extent, waste disposal problems in the industry [1].

## 4. Discussion

In this systematic review, 193 journal articles were screened and 47 articles that contained potential applications to valorise MAP residues were selected.

Existing reviews have focused, particularly, on potential sources and methodologies to recover phenolic compounds from distillation residues [28] and on the technology developed at laboratory scale and their application in industry to valorise residual biomass from MAP [1]. This is the first systematic review focusing specifically on the valorisation of all MAP processing residues, highlighting the associated benefits for sustainability. This study provides a set of future research topics, and provides a compilation of possibilities for decision-makers in the MAP industry to develop new businesses in the food, cosmetic, pharmaceutical, chemical, paper, or building industries using residues. Moreover, since the number of publications in this research field has been increasing, the more general and recent scope of this review allowed us to identify recent findings, as shown in Figure 4.

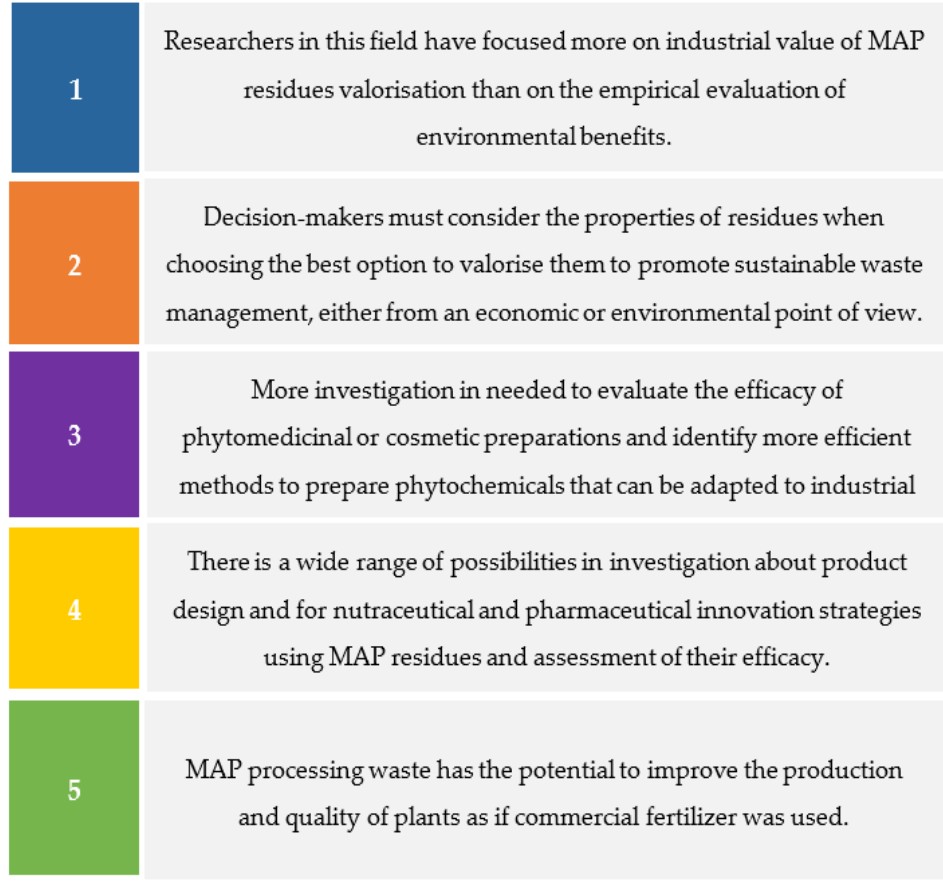

**Figure 4.** Findings' summarisation.

First, even though some articles refer generically to the positive effects of residue valorisation strategies on environmental sustainability, no studies were identified about an empirical evaluation of such effects in the extracted literature for this systematic review. Researchers in this field have focused more on the industrial value of MAP residues valorisation than on environmental benefits.

Threeat, the research aims of presenting waste management strategies that promote sustainability in the MAP sector lacks empirical evidence, which is a relevant limitation of this study.

Second, it was found that, to implement sustainable practices, decision-makers should prefer applications that consider the valuable properties of residues. For instance, if a specific type of residue is rich in phenolic compounds of interest, pyrolysis, gasification, and other applications may not be the optimal option to valorise them either from an economic or environmental point of view. The several MAP residues valorisation options mentioned in the literature are not equally sustainable. To implement circular economy strategies as a solution for more green waste management, it is necessary to evaluate the environmental impact associated. Additionally, no studies were identified about the economic feasibility assessment of MAP residues valorisation. The study by Saha and Basak [1] also stated that such an evaluation needs to be addressed in future research. Despite MAP residues having a great potential to be valorised, more studies about their application in real cases are needed.

Third, although several phytochemical studies point out the potential for valorisation of MAP residues, recent studies point to the need to evaluate the efficacy of phytomedicinal or cosmetic preparations [39] and identify more efficient methods to prepare phytochemicals that can be adapted to the industrial scale [50,51].

Fourth, a representative portion of the analysed literature presents phytochemical studies, suggesting that by-products from the MAP sector are a promising source of natural organic molecules for the development of innovative products in the field of biopesticides and biofertilisers, as well as in the food, pharmaceutical, and cosmetic industry. Therefore, there is a wide range of investigation possibilities for product design and nutraceutical and pharmaceutical innovation strategies using MAP residues and assessment of their efficacy.

Fifth, as demonstrated by Filipović et al. [34], MAP processing waste has the potential to improve the production and quality of plants as if commercial fertiliser were used. Replace chemical fertiliser usage with organic soil amendments reduces soil nutrient loss, sustains crop yields, and protects the environment [33], while reducing residues in MAP production. However, future research about the use of MAP residues as soil amendments is required to optimise the observed effects and to assess such applications in other crops and at the field-scale level [36]. Also, it is relevant to perform empirical investigations about the effects of the long-term application of MAP residues on productivity, soil quality, and greenhouse gas emissions [33]. These effects should also be discussed in comparison with the ones associated with the long-term application of chemical fertilisers.

Additionally, to encourage a progressive implementation of circular economy strategies to valorise MAP residues by the industry, further investigation is needed regarding:

- The identification of barriers and drivers for industrial implementation.
- The development of more efficient techniques and technology for the extraction of valuable compounds from MAP residues and for their usage to develop value-added products for crop management.
- The development of a decision support system to assess the sustainable performance considering economic, environmental, and social indicators, of both new and existing businesses based on MAP residues valorisation.

## 5. Conclusions

This systematic review allowed to conclude that residues from MAP processing can be used in:

1. Energy production through direct burning, making of briquettes after drying, pelletising and gasification or in biomethanisation and pyrolysis processes to obtain biodiesel and biochar, also for energy purposes.
2. Crop management. Solid residues can be submitted to composting, vermicomposting, anaerobic co-digestion, and anaerobic digestion. They can also be used to produce

biochar adsorbents by pyrolysis, as biosorbents, as soil organic cover or for weed management. Hydrolates can be valorised as eco-friendly biopesticides.

3. The chemical industry, namely in the production of anthraquinones, platform molecules, and enzyme synthesis.

4. The food industry, for both animal and human consumption. Their use in animal feed influenced *T. molitor* and aquaculture species' growth performance. Some by-products can be incorporated as supplement additives for the preservation of food.

5. The pharmaceutical and cosmetic industry, since they are valuable sources of phenolic compounds of interest, such as terpenes, phenolic monoterpenes, phenolic diterpenes, and flavonoids, among others.

6. Further applications such as the production of paper, the wood-based panel industry, composites synthesis, wastewater treatment, and building materials.

Thus, regarding practical implications, this research provides a comprehensive description of possible solutions to valorise MAP residues for more sustainable waste management practices that also offer new business opportunities.

Despite the wide range of possible applications, further investigation is needed to assess the effectiveness of circular economy approaches with MAP residues in real cases of industrial applications and economic viability.

**Author Contributions:** Conceptualisation, S.M., P.D.G. and A.P.; methodology, S.M., P.D.G. and A.P.; software, S.M.; validation, S.M., P.D.G. and A.P.; formal analysis, S.M.; investigation, S.M.; resources, S.M., P.D.G. and A.P.; data curation, S.M.; writing—original draft preparation, S.M.; writing—review and editing, S.M., P.D.G. and A.P.; visualisation, S.M., P.D.G. and A.P.; supervision, A.P. and P.D.G.; project administration, A.P. and P.D.G.; funding acquisition, A.P. and P.D.G. All authors have read and agreed to the published version of the manuscript.

**Funding:** This work was supported in part by the Fundação para a Ciência e Tecnologia (FCT), C-MAST (Centre for Mechanical and Aerospace Science and Technologies) under project UIDB/00151/2020 and under project POCI-01-0246-FEDER-181319 (PAM4Wellness), financed by the European Regional Development Fund (ERDF), of the European Union through POCI—Competitiveness and Internationalization Operational Program.

**Institutional Review Board Statement:** Not applicable.

**Informed Consent Statement:** Not applicable.

**Data Availability Statement:** Not applicable.

**Conflicts of Interest:** The authors declare no conflict of interest.

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
