# Peer review of "Sustainable Waste Management in the Production of Medicinal and Aromatic Plants—A Systematic Review"

_sustainability, doi:10.3390/su151813333_

Round 1

Reviewer 1 Report

Thanks for assigning the manuscript for review. I believe that the idea of this study has potential to be published, although I would recommend some changes in the manuscript. Suggestions to improve the quality of manuscript are as follows:

·       Add 3-4 highlights of the manuscript.

·       Highlights should summarize the research methodology and findings.

·       Focus on the novelty of the work while presenting highlights.

·       Keywords should be arranged in chronological order.

·       There is no flow in abstract and introduction section of the manuscript.

·       The objectives of the study should be clearly stated in the initial section. This will help the reader focus on the priorities.

·       Add the list of abbreviations.

·       The abstract suggests that further investigation is needed regarding the development of more efficient techniques and technology to develop value-added products and the assessment of sustainable performance. However, it does not provide concrete recommendations for future research or action. Including some specific suggestions for potential areas of further investigation would make the abstract more actionable and practical.

·       What research gaps does the current paper fill?

·       It mentions that circular economy is viewed as a solution for fostering a sustainable system planet and positively contributing to the economy and society. However, it does not elaborate on the specific benefits or advantages of circular economy approaches in the context of MAPs valorisation or waste management. Providing more specific examples or insights into how circular economy principles can address environmental risks and enhance economic opportunities would strengthen the abstract's argument.

·       Improve the scientific discussion on the long-term retention phase and compare the results with other studies.

·       The environmental risks associated with overexploitation and inadequate waste management of MAPs are increasing. However, the abstract does not specify what these environmental risks are or elaborate on their potential impacts. Providing some concrete examples or elaborating on the environmental risks would strengthen the abstract's argument.

·       Check Reference formatting. Check SUSTAINABILITY guidelines.

·       Please ensure that every reference cited in the text and the figure(s) is also in the reference list (and vice versa).

·       Please go through the following research articles for better understanding of the concerned topic:

https://doi.org/10.1016/j.chemosphere.2023.138579

https://doi.org/10.1016/j.envres.2022.113424

       https://doi.org/10.1016/j.jenvman.2021.113953

https://doi.org/10.1016/j.wasman.2021.11.044

       https://doi.org/10.1016/j.jenvman.2021.113953

https://doi.org/10.1016/j.chemosphere.2021.132451

      https://doi.org/10.1016/j.fuel.2022.127125.

The manuscript needs to be edited by a native english speaker

Author Response

Dear reviewer please see the document attached with the responses. 

Reviewer 2 Report

The article discusses the environmental risks arising from the increasing demand for Medicinal and Aromatic Plants (MAPs) and proposes the use of Circular Economy solutions to transform waste management issues into economic opportunities. A systematic review identifies multiple valorization possibilities for MAPs residues, including energy production and applications in various industries. 

I find the review well prepared and structured. The methods are appropriately stated and results are well-detailed. The discussion section is in support of the results.

The only comment I have is regarding Figure 1, where I would suggest the authors to define "reason 1" and "reason 2" in the figure legend.

Best wishes!

Author Response

Dear reviewer please see the document attached with the responses to all reviewers.

Reviewer 3 Report

Overall the manuscript gives detailed information of MAP's but lacks proper insight of management techniques as mentioned in the conclusion section. If the authors can add the techniques e.g., UASB, AD or else, that will give better readership to early career scientists. 

Minor English revision is required to improve scientific writing. 

Author Response

(The authors gave the same response as above.)

Reviewer 4 Report

Authors gave a birfe description and discussion on treatments of MAPs industry, which is obviously valuable information for both acdamic and industial circles. But authors mainly statemented the views or results from current literatures, which is a little bit not enough for a high quality review. Therefore, it would be better if some improved summary, new view or insights could be added.

Author Response

(The authors gave the same response as above.)

Reviewer 5 Report

General comment:

This paper presented a review study on sustainable waste management in the production of medicinal and aromatic plants. Although waste management research is an important field of study with much research interest, the authors failed to highlight the novelty of the study that differentiated the current study to similar studies available in publication. Much work is needed to improve the overall structure of the paper. For example, the novelty and contribution of the study are not adequately highlighted in the abstract. Hence, I conclude that the paper is not suitable for publication at its current stage. Hope below comments will able to help to further improve the paper.

Specific comment:

Abstract:

-      Language seems to be the main issue in this section as there are a lot of sentences that do not make sense.

-      For example, in the first sentence, how does the growing demand for MAPs related to inadequate waste management?

-      For logical explanation about the background of the study is needed before bringing in the idea of circular economy in the abstract.

-      Also, significance of the study and how the findings can be used to advance the field should be included.

-      An abstract is often presented separately from the article, so it must be able to stand alone.

-      Please try to merge all information into a paragraph with some attractive and new findings. The main result from the review is not seem in the abstract.

-      How does this review fill in any knowledge gap?

Introduction:

-      There are too many paragraphs. Try to merge similar points in a single paragraph. Carefully planning the key points.

-      Provide the statistics about the issues. Especially the effect of the waste management issue in terms of economy, the environmental effects etc.

-      Similarly, provide literature about the current state of waste management issue with MAPs, then highlight the shortcomings, leading to explanation about the knowledge gap.

-      Revised Introduction section based on the structure below:

1st paragraph: Problem statement

2nd paragraph: Current ongoing solution

3rd paragraph: Proposed solution in this work.

4th paragraph: Summarized the current research novelty and objective of this work.

-      Problem statement of your introduction is not strong, need to discuss more about it.

-      The earlier paragraphs should lead logically to specific objectives of the study.

-      Note that this part of the Introduction gives specific details: for instance, the earlier part of the Introduction may mention the importance of this study whereas the concluding part will specify what methods of control were used and how they were evaluated.

Materials and method:

-      What is the range of year for paper selection?

Results:

-      The introduction section about the subsections is unnecessary.

-      Most of the results are presented without critical evaluation.

-      For example: when reporting the highest number of articles by location, explain further about the importance of this findings, and how does it help to fill in research gap? Is it an information that is novel?

-      Similarly, more redaction is needed in the thematic results section.

-      The discussion is not in-depth to be considered a critical review.

-      Try to summarize the recent findings in an informative table.

-      The advances and bottleneck of current waste management of MAPs should be given more emphasis in the entire manuscript.

-      For each section/ subsection, it is advised to include the authors’ critical evaluation/ concluding remark.

-      The overall structure needs to be improved

-      Many long sentences, kindly break them down for better understanding

References

-      There are insufficient references in text

-      Some references are outdated. Kindly revise.

 Extensive editing of English language required

Author Response

(The authors gave the same response as above.)

Round 2

Reviewer 1 Report

Authors have complied the comments

Moderate editing is required.

Reviewer 3 Report

overall comments are addressed. the quality has improved. 

Reviewer 5 Report

The manuscript is corrected and revised according to the reviewer's comments. I am now satisfied with the new version, so I would like to recommend its publication.

Minor editing of English language required